# Silicon Nitride Whisker-Reinforced Aluminum Matrix Composites: Twinning and Precipitation Behavior

**Shoujiang Qu [1], Aihan Feng [1,*], Lin Geng [2], Jun Shen [1,3] and Daolun Chen [4,*]**

[1] School of Materials Science and Engineering, Tongji University, Shanghai 200092, China; qushoujiang@tongji.edu.cn (S.Q.); junshen@tongji.edu.cn (J.S.)

[2] School of Materials Science and Engineering, Harbin Institute of Technology, Harbin 150001, China; genglin@hit.edu.cn

[3] College of Mechatronics and Control Engineering, Shenzhen University, Shenzhen 518060, China

[4] Department of Mechanical and Industrial Engineering, Ryerson University, Toronto, ON M5B 2K3, Canada

\* Correspondence: aihanfeng@tongji.edu.cn (A.F.); dchen@ryerson.ca (D.C.); Tel.: 416-979-5000 (ext. 6487) (D.C.)

**Abstract:** Aluminum composites reinforced with ceramic whiskers exhibited a unique combination of high specific strength and superior specific modulus. A 20 vol.% $Si_3N_{4w}$/Al-11.5Si-1.0Mg-0.5Cu-0.5Ni (wt.%) composite was fabricated via squeeze casting in the present study. It was observed that the addition of silicon nitride ($Si_3N_4$) whiskers in the Al-Si cast alloy promoted extensive twinning in the eutectic silicon particles due to a coupled role of thermal stresses between the matrix and silicon and residual stresses present in the composite. Double aging peaks were present in the age-hardening curves. The precipitation mechanism involved the formation of $Mg_2Si$ and $Al_2CuMg$ phases. The presence of $Si_3N_4$ whiskers in the composite retarded the nucleation process of $Mg_2Si$ precipitate while enhancing its growth rate.

**Keywords:** composite; silicon nitride; Al-Si alloy; twinning; precipitation

## 1. Introduction

Aluminum matrix composites (AMCs) have been widely used for the lightweight structural applications in the aerospace, automotive, and other sectors because of their superior mechanical properties arising from the judicious combination and interaction between the matrix and reinforcement [1–5]. The matrix acts not only as a medium to transfer the load to reinforcements, but also as a load bearer [4,6]. There are different types of reinforcement, such as, SiCw, $Si_3N_4$w, carbon nanotubes (CNTs) [7,8], graphene nanoplatelets (GNPs) [9], and various ceramic particles in micro- or nano-sizes, etc. Thus, AMCs especially with precipitation-hardenable matrices possess superior strengths due to the added impediment to the motion of dislocations by precipitates [10]. Generally, the precipitation behavior of AMCs depends on the matrix material, type of reinforcements, processing route and aging temperature [11–14]. Previous studies indicated that the precipitation sequence in AMCs remained the same as the matrix alloys, but the aging kinetics was observed to be changed [5,15].

An accelerated aging response was reported in a number of composite systems, e.g., $B_4C$/Al-Zn-Mg alloy [15], $SiC_p$/2014Al [16], and $SiC_w$/Al-Li alloys [17,18]. A mechanism proposed to explain the accelerated aging in AMCs was related to the enhanced nucleation and/or growth of precipitates in the heavily dislocated matrix regions adjacent to the reinforcement [19,20]. The dislocation density in the composite (~$10^{13}$–$10^{14}$) was significantly (about 1~2 orders of magnitude) higher than that in the matrix alloys due to the coefficient of thermal expansion mismatch between the matrix and reinforcement [21–23]. Dislocations facilitated some precipitation due to the significantly decreasing

activation energy for nucleation [24]. Thus, the high density of dislocations served as heterogeneous nucleation sites, leading to the accelerated aging kinetics in the composites [25,26].

However, significant retardation in the aging kinetics of the composites has also been reported due to a reduction in the retained vacancies and the formation of interfacial phases [13,27–30]. The interfaces and dislocations could act as sinks for the vacancies, leading to a delayed formation of Guinier-Preston (GP) zones [31,32]. In addition, some studies indicated that there was a critical reinforcement size and volume fraction below which the aging behavior was unaltered [33,34].

Al-Si cast alloys exhibited superior wear resistance, strength, thermal conductivity, and low thermal expansion coefficient, and good casting characteristics [35–38]. The main alloying element silicon imparted high fluidity and low shrinkage, resulting in good castability and weldability [39–41]. Typical microstructures in the hypoeutectic Al-Si alloys consisted of primary aluminum solid solution ($\alpha$-Al) and Al-Si eutectic structure [42,43]. In the eutectic system where one phase solidifies in a faceted manner and the other nonfaceted, it was well known that the eutectic microstructure may change markedly with solidification conditions (cooling/growth rates) and also with minor additions of some modifying agents [44,45]. Semi-coherent interface between Al and Si phases with crystal orientation relationships, $[110]_{Al}//[110]_{Si}$ and $(111)_{Al}//(220)_{Si}$, indicated that the Si phase tended to grow along $(111)_{Al}$ plane [46]. Generally, Sr- and Na-based treatment was common practice for the modification of eutectic Si phase [47]. Mg and Cu were also important alloying elements in the Al-Si alloy, which greatly influenced aging kinetics. Cu/Mg and Mg/Si ratios affected the nature of various precipitates that form [48]. It was well accepted that two precipitation sequences were mainly responsible for the precipitation hardening of Al-Si-Cu/Al-Si-Mg alloys as follows [49]:

$$\alpha_{SSS} \rightarrow GP\ zone \rightarrow \beta'' \rightarrow \beta' \rightarrow \beta\ phase\ (Mg_2Si)$$
$$\alpha_{SSS} \rightarrow GP\ zone \rightarrow \theta'' \rightarrow \theta' \rightarrow \theta\ phase\ (Al_2Si)$$

where SSS stands for the supersaturated solid solution, $Al_2Cu$ and $Mg_2Si$ are two main strengtheners in the peak-aged condition [48]. Some other phases (e.g., S ($Al_2CuMg$) [50], and Q-$Al_5Cu_2Mg_8Si_6$ [40] or Q-$Al_3Cu_2Mg_9Si_7$ [51]) also existed in the aged Al-Si-Cu-Mg alloys. Double aging peaks in the Al-Si-Cu-Mg alloy were also reported by Li et al. [50].

In the AMCs the size, morphology and volume fraction of the reinforcement were important factors controlling the plasticity and residual stresses in the matrix. The sink effect due to the presence of interfaces between the matrix and reinforcement may also play a part in the precipitation kinetics. Our previous studies [42,52] indicated that the addition of $Si_3N_4$ whiskers accelerated the melting of the composite. The presence of $Si_3N_4$ whiskers led to a refinement of Al-Si eutectic structure and primary $\alpha$-Al dendrites. It also changed the morphology of primary $\alpha$-Al from dendrite-like to equiaxed, and the Al-Si eutectic structure from network-like to particulate [42]. However, it is unclear how the presence of $Si_3N_4$ whiskers affects the aging kinetics of matrix. Furthermore, while a number of researchers have reported twinning, i.e., the so-called impurity induced twinning (IIT) in the eutectic silicon phase of Al-Si cast alloys [44,53], no information on the twinning of $Si_3N_{4w}$/Al-Si composite is available in the literature so far. The present study was aimed to identify the effect of $Si_3N_4$ whiskers on the aging behavior of the composite, focusing on the Al-Si eutectic structure and twinning of eutectic silicon phase.

## 2. Materials and Methods

A 20 vol.% $Si_3N_{4w}$/4032Al composite was fabricated via squeeze casting at a pouring temperature of 800 °C. The composition of 4032Al matrix alloy was Al-11.5Si-1.0Mg-0.5Cu-0.5Ni with Fe ≤ 0.33 and Mn ≤ 0.15 (wt.%). The $\alpha$-$Si_3N_4$ whiskers had an average length of 20 μm and 0.1–1 μm in diameter. A preform with a size of Φ90 × 22 mm contained 103 g $\alpha$-$Si_3N_4$ whiskers. X-ray diffraction (XRD, Philips Xpert Panalytical ) and the subsequent JAD 6.5 software were used to identify the phases in the composite and the corresponding matrix alloy. Microstructure of the composite was examined via scanning electron microscope (SEM, HITACHI S-4700, Tokyo, Japan) equipped with energy-dispersive

sectroscopy (EDS). The samples for SEM observations were polished and some of them were etched by hydrofluoric acid. Differential scanning calorimeter (DSC, Netzsch STA449C, Germany) measurements were performed at a heating rate of 10 °C min$^{-1}$. The aging behavior was investigated by using Vickers hardness measurements, DSC and transmission electron microscopy (TEM, Philips-CM12). The TEM specimens were mechanically thinned to 50~60 μm, followed by ion-milling at 6 kV, 0.2 mA and an incident angle of 7~15°. Typically, a sample of 20 mg in the form of Φ4 mm disk was heated at 10 °C/min from 30 to 500 °C in a flowing argon atmosphere at a flow rate of 25 mL/min. DSC samples were solution heat-treated at 515 °C for 2 h in a salt bath, followed by quenching into water at room temperature. The Al-11.5Si-1.0Mg-0.5Cu-0.5Ni alloy and the composite were solution treated (typically at 515 °C for 40 min) and subsequently quenched in water. Aging was conducted at 175 °C for various dwelling times (1–28 h). The age-hardening response was determined by Vickers hardness measurements using HV-5 Vickers hardness tester at a load of 5 kg for 30 s at an aging time interval of 1 h. Each reported value was an average of at least five measurements.

## 3. Results and Discussion

Based on the Al-Si phase diagram, the basic microstructure was primary α-Al and Al-Si eutectic structure in the matrix Al-11.5Si-1.0Mg-0.5Cu-0.5Ni alloy. The Al-Si eutectic structure was observed to distribute along the primary α-Al grain boundaries [42]. The addition of Si$_3$N$_4$ whiskers led to a refined Al-Si eutectic structure and primary α-Al dendrites [52], since the morphology of Al-Si eutectic structure was changed from network-like to particulate [42]. It has also been shown that the silicon in the Al-Si cast alloys formed a continuous network, and twinning is an important branching mechanism [44,54–56].

Figure 1a is a secondary electron (SE) image showing typical microstructures of the Al-11.5Si-1.0Mg-0.5Cu-0.5Ni, and Figure 1b is a SE image showing typical microstructures of the Si$_3$N$_{4w}$/Al-11.5Si-1.0Mg-0.5Cu-0.5Ni composite. It is seen that the base alloy consisted of (i) larger α-Al primary phase and (ii) characteristic two-phase eutectic structure (eutectic α-Al and eutectic Si) surrounding the primary α-Al grains, i.e., network-like eutectic structure (Figure 1a). The addition of Si$_3$N$_4$ whiskers played a beneficial part in refining the primary α-Al grains and eutectic structure, with a microstructure where both whiskers and Si particles were fairly uniformly distributed in the Al matrix (Figure 1b,c). Figure 1d shows a backscattered electron (BSE) image of Si$_3$N$_{4w}$/Al-11.5Si-1.0Mg-0.5Cu-0.5Ni composite, with the results of EDS analyses listed in Table 1. It is seen that the grey phase in Figure 1d was eutectic Si particles, while the white phase was a FeMnNi-containing intermetallic compound. Such intermetallic compounds were also reported in other Al-Si cast alloys by Manasijević et al. [57] and Abdelaziz et al. [58].

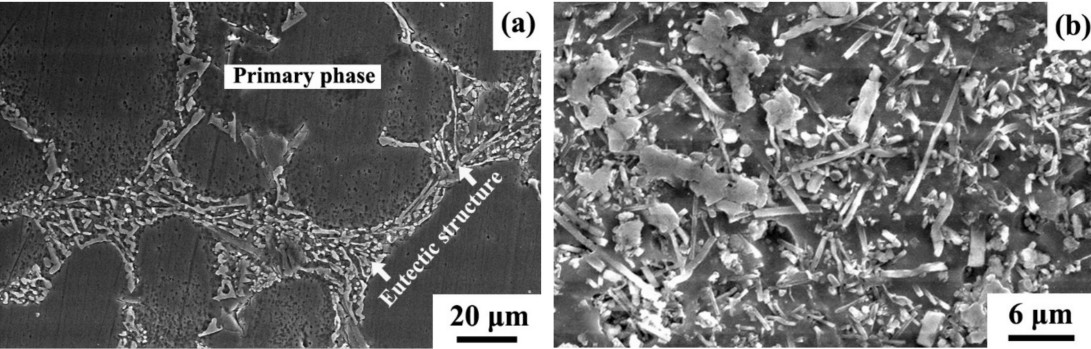

**Figure 1.** *Cont.*

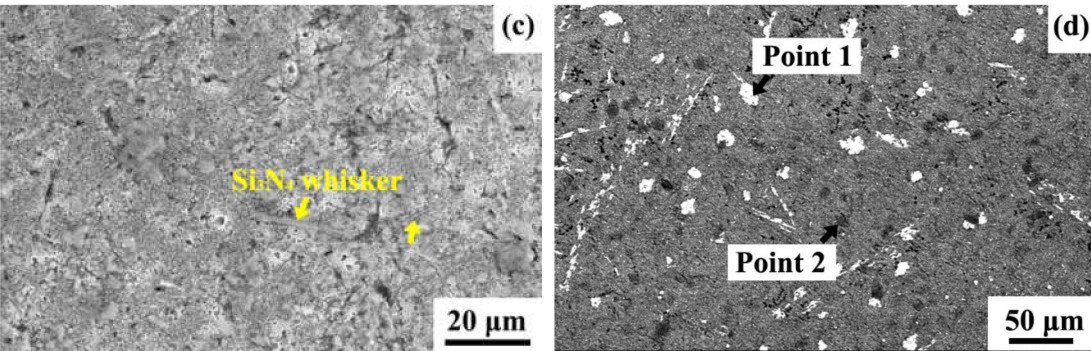

**Figure 1.** Typical SEM images showing the microstructures of Al-11.5Si-1.0Mg-0.5Cu-0.5Ni base alloy and Si$_3$N$_4$w/Al-11.5Si-1.0Mg-0.5Cu-0.5Ni composite: (**a**) SE image of the base alloy; (**b**) SE image of the composite; (**c**) secondary electron (SE) image and (**d**) backscattered electron (BSE) image of the composite at a lower magnification for an overall view.

**Table 1.** Results of EDS analyses of Si$_3$N$_4$w/Al-11.5Si-1.0Mg-0.5Cu-0.5Ni composite in Figure 1d.

| Composition (wt.%) | Al | Si | Mn | Fe | Ni |
|---|---|---|---|---|---|
| Point 1: White phase | 57.9 | 15.1 | 11.8 | 8.4 | 6.8 |
| Point 2: Grey phase | 3.3 | 96.7 | | | |

As shown in Figure 2a, in comparison with the matrix (or base) alloy, a large number of α-Si$_3$N$_4$ peaks were present, indicating that the current squeeze casting at a fairly high pouring temperature of 800 °C did not change the crystal structure of the α-Si$_3$N$_4$ whiskers added. The heat flow of the Al-11.5Si-1.0Mg-0.5Cu-0.5Ni alloy and Si$_3$N$_4$w/Al-11.5Si-1.0Mg-0.5Cu-0.5Ni composite, measured as a function of temperature, is shown in Figure 2b. Two endothermic peaks were observed in each curve, corresponding to the eutectic α-Al + Si peak (at a lower temperature) and primary α-Al solid solution peak (at a higher temperature), respectively. Obviously, the addition of Si$_3$N$_4$ whiskers moved ahead and accelerated the melting of the composite.

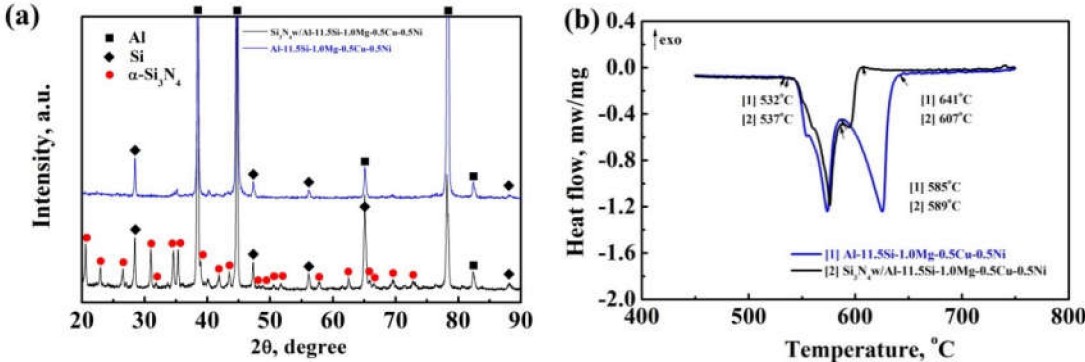

**Figure 2.** (**a**) X-ray diffraction (XRD) patterns and (**b**) DSC curves of Si$_3$N$_{4w}$/Al-11.5Si-1.0Mg-0.5Cu-0.5Ni composite and matrix, where DSC curves were obtained at a heating rate of 10 K/min.

Figure 3a shows that eutectic Si particles in the Si$_3$N$_{4w}$/Al-11.5Si-1.0Mg-0.5Cu-0.5Ni composite contained multiple twins, which could be better seen in a magnified image in Figure 3b. Figure 3c shows the corresponding selected area diffraction pattern (SADP) from both parent and twinned silicon parts, showing the formation of (111) twinning in the eutectic Si. The presence of a large number of twins suggests that Si$_3$N$_4$ whiskers could effectively promote twinning in eutectic silicon particles. This is in agreement with the observations in [59], where the number of twins formed in a compressed Mg-5Zn alloy increased when precipitate particles were present, due to the additional Orowan stresses

driving twin nucleation but inhibiting twin growth. Similar results were also reported in a ZEK110 (Mg-1Zn-1Gd-0.6Zr) alloy where the number density of twins nucleated was significantly high, although twin growth was retarded due to the presence of secondary phases [60]. Indeed, twinning in silicon has been observed, mainly (111) twins [61–64]. This is consistent with the present observations via SADP analysis in Figure 3c. In addition to the so-called impurity induced twinning (IIT), e.g., by strontium and cerium [44,53,65], the residual stresses or Orowan stresses caused by the difference in the coefficients of thermal expansion (ΔCTE) between the matrix and particles would play a major part in the composites. First, the huge ΔCTE difference between aluminum (~24 × 10$^{-6}$ (°C)$^{-1}$) and silicon (~2.6 × 10$^{-6}$ (°C)$^{-1}$) would lead to a potential residual strain (or mismatch strain) of as high as ~1.66%, estimated on the basis of thermal stress equation [$\Delta\varepsilon = \Delta\alpha_l \cdot \Delta T = (\alpha_{\mathrm{matrix}} - \alpha_{\mathrm{particle}})(T_{\mathrm{process}} - T_{\mathrm{RT}})$] assuming that the matrix and particles are well bonded, where $T_{\mathrm{process}}$ is the processing temperature (i.e., 800 °C in this study) and $T_{\mathrm{RT}}$ is room (or test) temperature (~25 °C). Even if the eutectic temperature of 577 °C in the Al-Si alloy system is used, instead of the processing/pouring temperature of 800 °C, the estimated mismatch strain is still ~1.18%. This was equivalent to exert a strong restraint pressure on the silicon particles, causing the occurrence of twinning. The presence of Si$_3$N$_4$ whiskers would further enhance the twinning of silicon particles, since the residual stresses or Orowan stresses present in the matrix of a composite stemming from the existence of ceramic particles can play an added role in pressing the silicon particles, thus promoting the formation of multiple twins in the silicon phase, as shown in Figure 3.

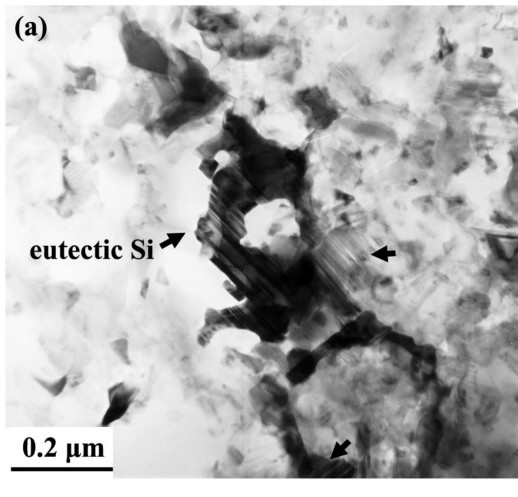

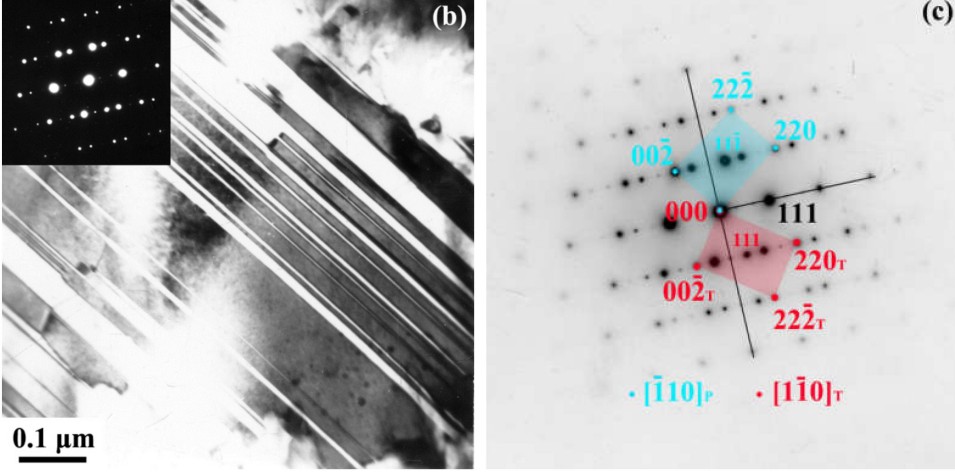

**Figure 3.** TEM images showing eutectic Si in the Si$_3$N$_4$w/Al-11.5Si-1.0Mg-0.5Cu-0.5Ni composite: (**a**) Eutectic Si particles; (**b**) Twinning in the eutectic Si; and (**c**) The corresponding selected area diffraction pattern (SADP) in Figure 1b.

Figure 4a shows the change of Vickers hardness of the composite and the corresponding matrix alloy with time on a semi-logarithmic scale during artificial aging. The enhanced hardness generated by the addition of $Si_3N_4$ whiskers was significant. The double aging peaks were observed to be present in both the composite and matrix alloy. For the composite, the first and second peak appeared after aging for 5 and 12 h, respectively. The peak hardness was due to co-precipitation of $Mg_2Si$ and $Al_2CuMg$ phases, which could be revealed via DSC as suggested by Charai et al. [66]. Typical DSC curves are shown in Figure 4b, where two distinct exothermic peaks A and B were visible in the composite and matrix alloy, respectively. One was due to the formation of two kinds of GP II zones, and the other due to two kinds of metastable precipitates (Figure 4b). The first exothermic peak appeared when the temperature exceeded 240 °C, and the second peak occurred at ~285 °C in the matrix alloy. The two precipitation exothermic peaks of the composite were partially overlapped at 286 °C and 306 °C. It can also be seen that the heat flow for peak A was higher than that for peak B in both composite and matrix alloy. Two exothermic peaks in the matrix alloy were significantly sharper than those in the composite. The transition from GP II to metastable phases in DSC curve, i.e., the dissolution of GP zones and the nucleation of metastable phases on dislocations, may be the main reason for the formation of double aging peaks [50]. It is clear that the DSC results in Figure 4b corresponded well to the hardness changes in Figure 4a.

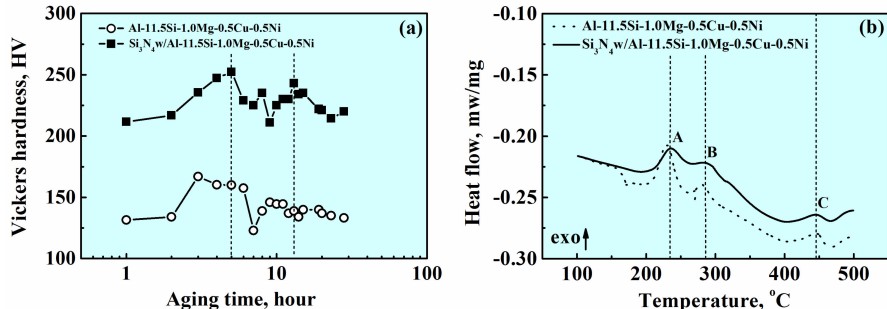

**Figure 4.** (**a**) Hardness-time curves during age-hardening; and (**b**) Differential scanning calorimeter (DSC) curves for the Al-11.5Si-1.0Mg-0.5Cu-0.5Ni matrix alloy and $Si_3N_4w$/Al-11.5Si-1.0Mg-0.5Cu-0.5Ni composite.

The high density of dislocations generated at the $Si_3N_{4w}$/Al-11.5Si-1.0Mg-0.5Cu-0.5Ni interfaces can be seen from Figure 5a, indicating the presence of residual stresses in the composite and corroborating the higher hardness in Figure 4a, while Figure 5b,c show the precipitates of the matrix in the composite after aging for 5 and 6 h, respectively. The TEM images uncovered two types of precipitates: needle-shaped precipitates and lamellar phase, which were homogenously distributed in the matrix (Figure 5c), and revealed via SADP (Figure 5d) to be β-$Mg_2Si$ and S-$Al_2CuMg$ phase, respectively. The needles were seen to have coarsened into rods. These coarse rod-like precipitates were the metastable β′ phase. A lot of fine needle-shaped precipitates, determined to be β phase by the SADP, could also be seen in the matrix alloy (Figure 6). The quantity of precipitates was larger in the matrix alloy, in comparison with the composite.

The addition of $Si_3N_4$ whiskers directly influenced the aging kinetics by the segregation of the solute atoms at the matrix/reinforcement interface. Furthermore, the addition of $Si_3N_4$ whiskers modified the precipitate nucleation condition associated with dislocation density and vacancy concentration, resulting in accelerating or delaying precipitation kinetics. A high dislocation density and the interfacial stresses affected not only the heterogeneous nucleation of precipitates, but also served as short-circuit paths for pipe diffusion which could accelerate the aging process. Dislocations are known to be sites for heterogeneous nucleation and paths for increased atomic transport during growth [19]. However, the $Mg_2Si$ phase basically nucleated on vacancy [19]. First, the high dislocation density in the composite retarded the nucleation process of $Mg_2Si$ precipitates by the reduction of the excess vacancy concentration in the matrix due to the absorption of vacancies into the interfaces between

the whisker and matrix, or at dislocations. The screw dislocations absorbed the vacancies to form the coil dislocations, and the edge dislocations absorbed the vacancies to form dislocation jogs. Second, the high dislocation density may also enhance $Mg_2Si$ precipitate growth rate by serving as short-circuit paths for solute diffusion [11]. The enhanced diffusivity of Mg and Cu atoms would therefore increase the growth rate of $Mg_2Si$ phase, resulting in a larger size of $Mg_2Si$ precipitates with a lower number density in the composite (Figure 5c) than that in the matrix alloy (Figure 6a).

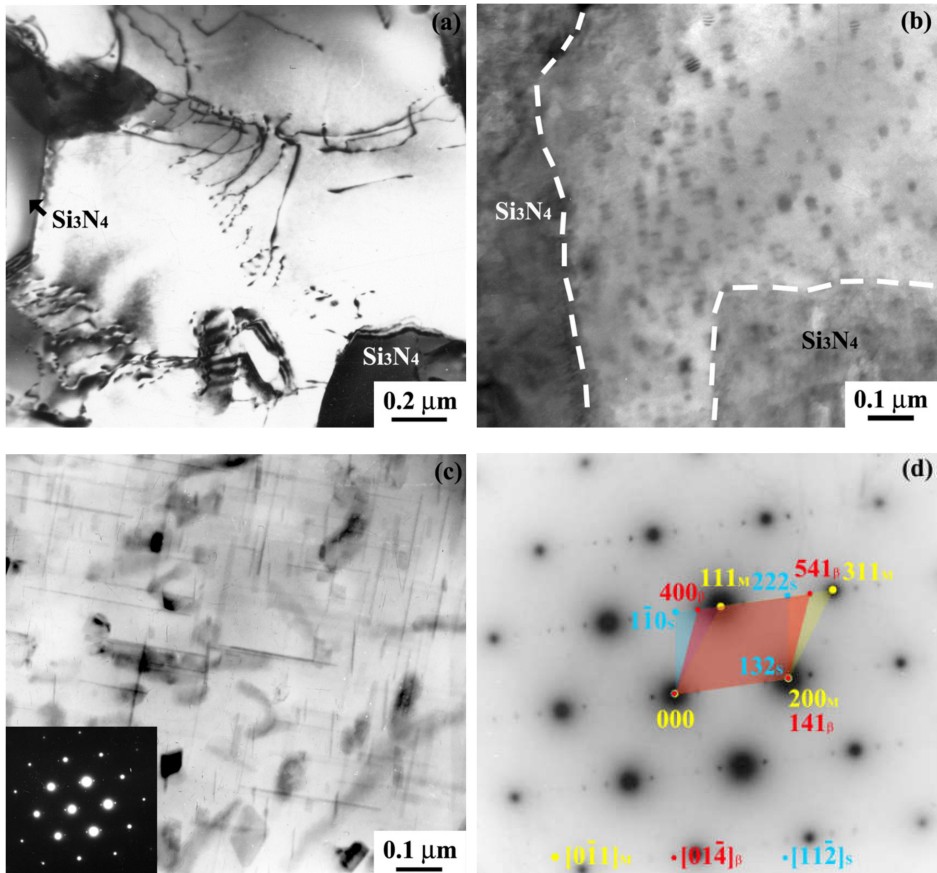

**Figure 5.** Transmission electron microscopy (TEM) micrographs of $Si_3N_4w/Al$-11.5Si-1.0Mg-0.5Cu-0.5Ni composite: (**a**) As-cast composite; (**b**) Composite after aging for 5 h; (**c**) Composite after aging for 6 h; and (**d**) The corresponding SADP of matrix, S-$Al_2CuMg$ and β-$Mg_2Si$ phases in (**c**).

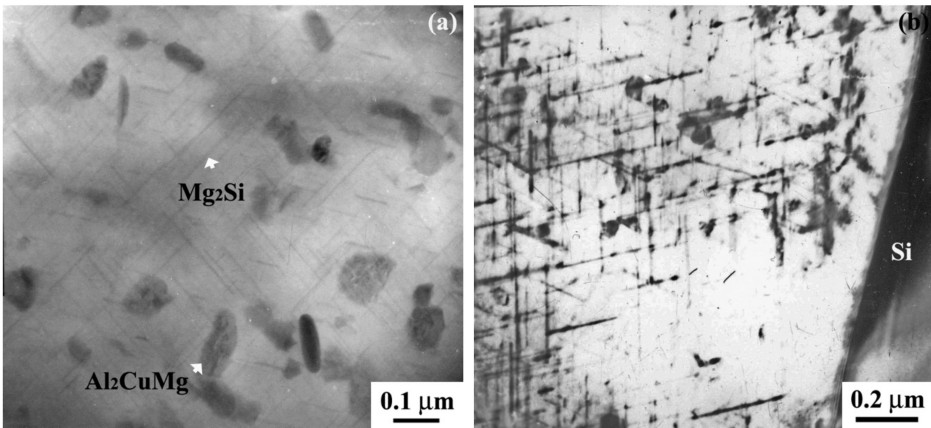

**Figure 6.** TEM images of Al-11.5Si-1.0Mg-0.5Cu-0.5Ni alloy after aging for 6 h: (**a**) A bright field image; and (**b**) Precipitates near Si phase.

## 4. Conclusions

(1) The addition of $Si_3N_4$ whiskers in the Al-11.5Si-1.0Mg-0.5Cu-0.5Ni alloy played an important role in refining primary $\alpha$-Al grains and eutectic structure, where both whiskers and Si particles were fairly uniformly distributed in the Al matrix.

(2) The presence of $Si_3N_4$ whiskers promoted multiple twinning in the eutectic silicon of the $Si_3N_{4w}$/Al-11.5Si-1.0Mg-0.5Cu-0.5Ni composite.

(3) Double aging peaks were present in the aging hardening curves of the composite and the corresponding matrix alloy. The precipitation mechanism involved the formation of $\beta$-$Mg_2Si$ and S-$Al_2CuMg$ phases in both composite and matrix alloy.

(4) The added $Si_3N_4$ whiskers retarded the nucleation process of $Mg_2Si$ precipitates, while accelerating the growth of $Mg_2Si$ precipitates.

**Author Contributions:** Conceptualization, S.Q., L.G., and D.C.; methodology, A.F. and J.S.; investigation, S.Q. and A.F.; writing—original draft preparation, S.Q.; validation, S.Q. and J.S.; writing—review and editing, A.F. and D.C.; project administration, L.G. and D.C. All authors have read and agreed to the published version of the manuscript.

**Funding:** The authors are grateful for the finance support from the National Nature Science Foundation of China (Grant No.50071018) and Natural Sciences and Engineering Research Council of Canada (NSERC) in the form of international collaboration.

**Conflicts of Interest:** The authors declare no conflict of interest.

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
