# Peer review of "Silicon Nitride Whisker-Reinforced Aluminum Matrix Composites: Twinning and Precipitation Behavior"

_metals, doi:10.3390/met10030420_

Round 1

Reviewer 1 Report

Our previous studies [39,49] indicated that the addition of Si3N4 whiskers accelerated the 75 melting of the composite. Please explain wider (1 sentence) the reason of that effect.

Please indicate why the temperature of 175oC was chosen in your experiments for the artificial ageing.

In the chapter discussion, your opinion on practical aspects of obtained results is  expected (two peaks in hardness curves). If your heat treatment parameters can be recommended as a final process parameters? What do you expect in processes of composite natural ageing  (role of whiskers similaqr  or not)?

Author Response

Response to the reviewer comments

Editorial reference number: Metals-736209

The authors would like to sincerely thank the Reviewer and the Editor for taking the time to review our manuscript and to provide us with constructive feedback and excellent suggestions for improving our paper. We have addressed all the comments as best we could.

Our response to the individual comments is provided below. The reviewer’s comments are in bold (italic), and our replies are in a regular font. The changes have been yellow-colored in the text.

Response to Reviewer 1 Comments

Point 1-1: Our previous studies [39,49] indicated that the addition of Si3N4 whiskers accelerated the 75 melting of the composite. Please explain wider (1 sentence) the reason of that effect.

Response 1-1: Thanks for the reviewer’s advice. We have added Figure 2b and the related description to explain the reason.

Point 1-2: Please indicate why the temperature of 175oC was chosen in your experiments for the artificial ageing.

Response 1-2: Thanks for the reviewer’s advice. We chose the aging temperature of 175°C based on references and our earlier experiments.

Point 1-3: In the chapter discussion, your opinion on practical aspects of obtained results is expected (two peaks in hardness curves). If your heat treatment parameters can be recommended as a final process parameters? What do you expect in processes of composite natural ageing  (role of whiskers similaqr  or not)?

Response 1-3: Thanks for the reviewer’s comment. In Fig. 4a, the peak aging is 5 h for the composite, while 3h for the matrix. The addition of Si3N4 whiskers directly influenced the aging kinetics.

Reviewer 2 Report

This study investigated the twinning and precipitation behavior in silicon nitride whisker-reinforced aluminum-silicon composites. The fabricated materials were well-characterised by HRTEM which supported the results elucidated from this study. However, following comments should be dealt before the publication of this article.

Abstract:

Authors provided a very short abstract and significance and novelty of this work has not been stated. Abstract should be revised to reflect the significance of this study. Also correct the typos in keywords.

Introduction:

  • Line 27: ‘mechanical properties due to synergetic effects between the matrix and reinforcement’ ..What do you mean by synergetic effects between matrix and reinforcements? Or it may be superior mechanical properties due to underlying synergetic strengthening mechanisms in these kinds of materials.
  • Lines 47-48: interfaces and dislocations could act as sinks for the vacancies, leading to a delayed formation of GP zones [48] ..what do you mean by GP zones? If using an abbreviation for the first time, provide full explanation of this first.
  • Discuss briefly different kind of other reinforcement materials which have been used in AMCs such as CNTs, GNPs etc.

Materials and Methods:

  • Lines 87-88: ‘The -Si3N4 whiskers had an average length of 20 m and 0.1-1 87 m in diameter. A preform with a size of Φ90×22 mm contained 103 g -Si3N4 whiskers’ . Correct the mistakes in this important information about the raw materials used in this study and also provide the sources (supplier information).
  • Line 91: ‘TEM were prepared by ion-milling technique’. Provide full explanation of TEM sample preparation. This is an important information as phases, twins, etc may also form during sample preparation such as grinding, dimpling, and then milling procedures during TEM sample preparation.
  • Line 96: Artificial aging was conducted at 175C for various dwelling times (1-28 h). What do you mean by artificial aging? Use scientific language here for the readers.
  • Lines 97-98: ‘Vickers hardness measurements using Leco-LM-247AT hardness tester 97 at a load of 5 kg for 30 s at an interval of 1 h.’ Are you sure that you applied 5 kg load for hardness measurements? Verify.

Results:

  • Provide a detailed microstructural analysis by SEM/EDX and XRD studies. This information is important and should be included in this article.
  • Where are the images of raw materials? SEM images should be provided for whiskers used in this study.

Conclusions:

  • Authors claimed that the addition of Si3N4 whiskers refined the Al-Si eutectic structure but there is no information provided to claim grain refinement in fabricated composites by addition of Si3N4 whiskers.
  • Authors claimed that the precipitation mechanism involved the formation of β-Mg2Si 203 and S-Al2CuMg phases in both composite and matrix alloy. However, not much information is provided to see these phases in bulk composites. XRD should be done on these materials as TEM characterization is not enough. XRD results can compliment TEM studies so both studies should be provided in the article.

Author Response

Response to the reviewer comments

Editorial reference number: Metals-736209

The authors would like to sincerely thank the Reviewer and the Editor for taking the time to review our manuscript and to provide us with constructive feedback and excellent suggestions for improving our paper. We have addressed all the comments as best we could.

Our response to the individual comments is provided below. The reviewer’s comments are in bold (italic), and our replies are in a regular font. The changes have been yellow-colored in the text.

Response to Reviewer 2 Comments

Point 2-1: This study investigated the twinning and precipitation behavior in silicon nitride whisker-reinforced aluminum-silicon composites. The fabricated materials were well-characterised by HRTEM which supported the results elucidated from this study. However, following comments should be dealt before the publication of this article.

Abstract: Authors provided a very short abstract and significance and novelty of this work has not been stated. Abstract should be revised to reflect the significance of this study. Also correct the typos in keywords.

Response 2-1: Thanks for the reviewer’s advice. We have revised and expanded the abstract, and corrected the typos in keywords.

Point 2-2: Introduction: Line 27: ‘mechanical properties due to synergetic effects between the matrix and reinforcement’ ..What do you mean by synergetic effects between matrix and reinforcements? Or it may be superior mechanical properties due to underlying synergetic strengthening mechanisms in these kinds of materials.

Response 2-2: Thanks for the reviewer’s comment. This sentence has been re-worded.

Point 2-3: Lines 47-48: interfaces and dislocations could act as sinks for the vacancies, leading to a delayed formation of GP zones [48] ..what do you mean by GP zones? If using an abbreviation for the first time, provide full explanation of this first.

Response 2-3: Thanks for the reviewer’s suggestion. We have added the full name of Guinier-Preston for the GP zones in the text.

Point 2-4: Discuss briefly different kind of other reinforcement materials which have been used in AMCs such as CNTs, GNPs etc.

Response 2-4: Thanks for the reviewer’s advice. We have added this information in the introduction.

Point 2-5: Materials and Methods: Lines 87-88: ‘The -Si3N4 whiskers had an average length of 20 m and 0.1-1 87 m in diameter. A preform with a size of Φ90×22 mm contained 103 g -Si3N4 whiskers’ . Correct the mistakes in this important information about the raw materials used in this study and also provide the sources (supplier information).

Response 2-5: Thanks for the reviewer’s comment. We have revised it.

Point 2-6: Line 91: ‘TEM were prepared by ion-milling technique’. Provide full explanation of TEM sample preparation. This is an important information as phases, twins, etc may also form during sample preparation such as grinding, dimpling, and then milling procedures during TEM sample preparation.

Response 2-6: Thanks for the reviewer’s comment. We have more information in the text.

Point 2-7: Line 96: Artificial aging was conducted at 175°C for various dwelling times (1-28 h). What do you mean by artificial aging? Use scientific language here for the readers.

Response 2-7: Thanks for the reviewer’s comment. We have deleted the word “artificial”.

Point 2-8: Lines 97-98: ‘Vickers hardness measurements using Leco-LM-247AT hardness tester 97 at a load of 5 kg for 30 s at an interval of 1 h.’ Are you sure that you applied 5 kg load for hardness measurements? Verify.

Response 2-8: Thanks for the reviewer’s comment. We checked the Vickers hardness measurements. We used a normal Vickers hardness tester (not a Vickers microhardneness tester) at a load of 5 kg for 30 s at an aging time interval of 1h.

Point 2-9: Results: Provide a detailed microstructural analysis by SEM/EDX and XRD studies. This information is important and should be included in this article.

Response 2-9: Thanks for the reviewer’s advice. We have added Figures 1 and 2 and the related description.

Point 2-10: Where are the images of raw materials? SEM images should be provided for whiskers used in this study.

Response 2-10: Thanks for the reviewer’s comment. We have added Figure 1 showing the whiskers of raw materials.

Point 2-11: Conclusions: Authors claimed that the addition of Si3N4 whiskers refined the Al-Si eutectic structure but there is no information provided to claim grain refinement in fabricated composites by addition of Si3N4 whiskers.

Response 2-11: Thanks for the reviewer’s comment. As suggested by the reviewer, two more figures (Figures 1 and 2) and the relevant description have been added to support this.

Point 2-12: Authors claimed that the precipitation mechanism involved the formation of β-Mg2Si203 and S-Al2CuMg phases in both composite and matrix alloy. However, not much information is provided to see these phases in bulk composites. XRD should be done on these materials as TEM characterization is not enough. XRD results can compliment TEM studies so both studies should be provided in the article.

Response 2-12: Thanks for the reviewer’s excellent advice. XRD results could verify the TEM studies well. This would be our further study when our lab is re-opened after the current novel coronavirus crisis is over.

Reviewer 3 Report

The article can be accepted after making some corrections.
1. The abstract should be extended in such a way that it fully reflects the content of the article

2. Some letters are missing from the Materials and method section. In addition, the Si3N4 powder manufacturer and its purity must be provided.

3. The manuscript should be checked for grammar.

Author Response

Response to the reviewer comments

Editorial reference number: Metals-736209

The authors would like to sincerely thank the Reviewer and the Editor for taking the time to review our manuscript and to provide us with constructive feedback and excellent suggestions for improving our paper. We have addressed all the comments as best we could.

Our response to the individual comments is provided below. The reviewer’s comments are in bold (italic), and our replies are in a regular font. The changes have been yellow-colored in the text.

Response to Reviewer 3 Comments

Point 3-1: The article can be accepted after making some corrections. The abstract should be extended in such a way that it fully reflects the content of the article.

Response 3-1: Thanks for the reviewer’s advice. We have revised and extended the abstract.

Point 3-2: Some letters are missing from the Materials and method section. In addition, the Si3N4 powder manufacturer and its purity must be provided.

Response 3-2: Thanks for the reviewer’s advice. We have revised it.

Point 3-3: The manuscript should be checked for grammar.

Response 3-3: Thanks for the reviewer’s advice. We have revised and checked grammar.

Round 2

Reviewer 2 Report

Authors have now addressed all the reviewers' comments and therefore should be accepted for publication in its present form.